# Infectious Risk in Pediatric Emergency Departments in Italy: A Survey by the Italian Society for Pediatric Emergency and Urgent Medicine (SIMEUP) on Available Preventive and Diagnostic Tools

**DOI:** 10.3390/jcm13247762

**Published:** 2024-12-19

**Authors:** Sonia Bianchini, Stefania Formicola, Lidia Decembrino, Laura Ladetto, Maria Novella Pullano, Cosimo Neglia, Danilo Buonsenso, Stefania Zampogna, Susanna Esposito

**Affiliations:** 1Pediatric Unit, ASST Santi Carlo e Paolo, 20153 Milan, Italy; bianchini.sonia@outlook.it; 2Pneumologia and UTSIR Unit, Santobono Pausilipon Hospital, 80129 Naples, Italy; stefaniaformicola10@gmail.com; 3Pediatric and Neonatology Unit, 27029 Vigevano, Italy; lidiadec26@gmail.com; 4Pediatric and Neonatology Unit, ASL TO4, 10036 Turin, Italy; laura.ladetto@gmail.com; 5Pediatric Unit, Ospedale Pugliese Ciaccio, 88100 Catanzaro, Italy; novellapullano@gmail.com; 6Pediatric Clinic, Department of Medicine and Surgery, University of Parma, Via Gramsci, 14, 43125 Parma, Italy; negliamino@gmail.com; 7Department of Woman and Child Health and Public Health, Fondazione Policlinico Universitario Agostino Gemelli-IRCCS, 00168 Rome, Italy; danilo.buonsenso@policlinicogemelli.it; 8Università Cattolica del Sacro Cuore, 00168 Rome, Italy; 9Pediatric Unit, Crotone Hospital, 88837 Crotone, Italy; stezampogna@gmail.com

**Keywords:** pediatric emergency room, infection control, COVID-19, rapid diagnostic tests, hospital preparedness

## Abstract

**Background/Objectives:** The COVID-19 pandemic has emphasized the importance of preparedness in preventing the spread of infectious diseases, especially in Emergency Departments (EDs), where initial patient assessments and triage occur. This study aims to evaluate the current practices and available tools for infection control in Pediatric EDs across Italy, focusing on the differences between various hospital types and regional settings. **Methods:** A cross-sectional national survey was conducted in February 2022, targeting healthcare workers in Pediatric EDs across Italy. The survey, distributed via the Italian Society for Pediatric Emergency and Urgent Medicine (SIMEUP) mailing list, collected data on infection control measures, including the availability of hand hygiene stations, personal protective equipment, disinfection protocols, and the use of rapid diagnostic tests. **Results:** A total of 80 questionnaires were completed from 119 (67.2%) different ERs. The majority of respondents were from Northern Italy (47.5%) and worked in hospitals with 24 h pediatric assistance (48.8%). Less than half of non-pediatric hospitals had separate access for children, potentially exposing them to adult pathogens. Across all settings, basic infection control measures, such as providing masks and hand gel, were widely implemented. However, significant differences were observed in the availability of social distancing, informational materials, and dedicated pediatric pathways, with I level assistance hospitals less likely to have these resources. Rapid diagnostic tests were available in most settings, but the focus was predominantly on SARS-CoV-2, despite other respiratory pathogens’ relevance in pediatric care. **Conclusions:** Strengthening preparations for future pandemics will be crucial in enhancing the resilience of healthcare systems and ensuring the safety of both patients and healthcare workers in the face of emerging infectious threats.

## 1. Introduction

The COVID-19 pandemic has underscored the critical importance of preparedness in managing new and recurring infectious diseases, emphasizing the need for robust prevention strategies to mitigate their spread [1,2,3]. The Emergency Department (ED) serves as the initial point of contact for patients, where they are assessed, triaged, and directed to appropriate wards for hospitalization or further examination. While healthcare systems worldwide have made efforts to prepare for potential surges in acute care patients, there remains a knowledge gap regarding how well these facilities have trained their staff for such crises.

A study by Hartford and colleagues documented the response to the pandemic at two academic children’s hospitals in the United States, detailing the processes of access, triage, testing, and patient segregation. They also highlighted the flexibility and resilience required of healthcare workers as they adapted to rapidly evolving workflows and guidelines during the pandemic [4]. This study reflects broader challenges faced by EDs globally, where previous infectious disease outbreaks and natural disasters have disrupted clinical operations in various ways, often contingent on the nature and duration of the crisis [5].

The literature reveals varying levels of preparedness in EDs to handle such events, with some reports highlighting significant deficiencies [6,7,8]. The World Health Organization (WHO) has issued comprehensive guidelines to assist healthcare systems in managing emergency situations, but their implementation remains inconsistent [9]. Bressan et al. conducted a survey across European pediatric EDs, identifying significant gaps in preparedness and responses to COVID-19. These included the lack of early availability of documented contingency plans, insufficient simulation training, inappropriate use of personal protective equipment (PPE), and inadequate isolation facilities, all of which are critical to overcoming future pandemics [10].

Seasonal epidemics caused by pathogens such as respiratory syncytial virus (RSV) and rotavirus highlight the persistent challenge of infectious disease transmission in healthcare settings, particularly in pediatric Eds [11,12,13,14]. These pathogens, like SARS-CoV-2, are highly transmissible, particularly in young populations, and their seasonal surges impose significant strain on healthcare systems. RSV, for example, is a leading cause of bronchiolitis and pneumonia in infants, often resulting in hospitalizations during the colder months [11,12]. Similarly, rotavirus remains a primary contributor to severe gastroenteritis in children under five, despite the availability of vaccines [13,14].

Despite these insights, there is limited information on how triage is specifically conducted during infectious disease outbreaks in pediatric settings. Additionally, there is a dearth of data on the cleaning protocols and the availability of tools within ERs aimed at reducing infection transmission. Given these gaps, we conducted this study to investigate current practices in the prevention of infectious disease transmission in Italian Pediatric EDs. Our focus is on the protocols, tools, and information available to both healthcare workers and patients, with the aim of identifying areas for improvement.

## 2. Methods

To assess the preparedness and practices in preventing infectious disease transmission in Pediatric EDs across Italy, a comprehensive survey was designed and administered by the Infectious Group of the Italian Society for Pediatric Emergency and Urgent Medicine (SIMEUP). The study utilized a cross-sectional multicenter national survey, targeting healthcare workers within Pediatric EDs.

### 2.1. Survey Design and Distribution

The survey was meticulously crafted to capture a broad range of data, including demographic information, current practices in infection prevention, availability of diagnostic tools, adherence to protocols, and the perceived adequacy of training related to infectious disease management. The survey instrument was carefully designed to capture a comprehensive range of data on infection prevention practices in pediatric EDs. To ensure survey reliability and validity, the instrument was developed in collaboration with subject matter experts, incorporating iterative revisions to enhance clarity and relevance, while pre-testing with a subset of respondents was conducted to identify and rectify potential ambiguities, thereby supporting the consistency and accuracy of the collected data. The final survey (Appendix A) was constructed using Google Forms, a platform chosen for its ease of distribution and data collection.

The target population for this survey consisted of healthcare professionals working in pediatric EDs across Italy (i.e., the Director of the Unit of each center), as identified through the SIMEUP mailing list. In February 2022, the survey was disseminated through the SIMEUP mailing list, reaching healthcare workers across various Pediatric EDs in Italy. To maximize response rates, three reminder emails were sent at weekly intervals to those who had not responded promptly. This strategy was employed to encourage participation and ensure a representative sample of the target population.

### 2.2. Data Collection and Management

All responses were automatically recorded in Google Forms and subsequently exported to Microsoft Excel^®^ for data management and initial processing. The data were carefully reviewed for completeness and accuracy. Any incomplete or inconsistent responses were excluded from the final analysis.

### 2.3. Statistical Analysis

Data analysis was conducted using STATA^®^ Software (Release 12, College Station, TX, USA). The analysis included both descriptive and inferential statistics. For descriptive analysis, the absolute frequency and percentage were reported for dichotomous and categorical variables, providing an overview of the distribution of responses across different categories.

To examine associations between variables, comparisons were made using the chi-square test when the expected frequency was greater than 5. For variables with an expected frequency of less than 5, Fisher’s exact test was employed to ensure the accuracy of the results. A *p*-value of less than 0.05 (*p* < 0.05) was considered statistically significant, indicating a meaningful association between the variables under investigation.

## 3. Results

A total of 80 questionnaires were collected from healthcare professionals across 119 (67.2%) different EDs in Italy. Table 1 summarizes the main characteristics of the respondents and their respective centers. The response rate was consistent across regions, ensuring a balanced representation of pediatric EDs from the North, Center, South, and Isles of Italy. The majority of respondents were from the North of Italy (47.5%), were over 35 years old (88.7%), and were primarily medical doctors (81.3%). The types of hospitals represented, in order of frequency, were as follows: hospitals with 24 h pediatric assistance (48.8%), II level EDs (25.0%), I level EDs (11.3%), pediatric hospitals (8.8%), and Institutes of Scientific Hospitalization and Care (IRCCS) (6.3%). Among hospitals that were not specifically pediatric, 42.5% had separate access for children in the ED, and more than half (60.3%) had a fast track for pediatric patients.

In all the hospitals, a lack of planning, insufficient simulation training, inappropriate use of PPE, and inadequate isolation facilities were reported prior to the beginning of the pandemic. Table 2 compares the differences in infection control practices across three hospital settings: first level of assistance (including first level of assistance and I level EDs), second level of assistance (24 h pediatric assistance hospitals and II level EDs), and third level of assistance (pediatric hospitals and IRCCSs). A specific infectious disease triage was most commonly found in third level assistance hospitals (75%), followed by second-level (50%) and first-level (41.67%) hospitals, though these differences were not statistically significant.

In all hospital settings, masks and hand sanitizers were provided before entering the visit room. First-level assistance hospitals were more likely to ensure social distancing (64.58%) and provide informational materials about infectious diseases (43.75%). Protective equipment was universally worn by healthcare workers in all settings, and sinks were available in nearly all visit rooms.

First-level assistance hospitals were more likely to have single rooms for short-stay observation (35.42% vs. 16.67% in third-level hospitals), with a higher prevalence of private bathrooms in these rooms (83.33% vs. 33.33% in third level hospitals, *p* < 0.01).

Disinfection protocols were consistently present across all hospital settings (Table 3), with more than two thirds of settings performing disinfection at least three times per day (64.58% in first-level, 60% in second-level, and 75% in third-level hospitals). However, there was variability in the handling of toys, with many hospitals not providing toys at all (75% in third-level, 35% in second-level, and 45.83% in first-level hospitals).

Rapid diagnostic tests, including urine sticks, blood gas analysis, and SARS-CoV-2 antigen tests, were widely available across all settings (Table 4). However, the location and personnel responsible for conducting these tests varied. In first- and second-level assistance hospitals, nurses primarily performed the tests, whereas in third-level hospitals, doctors were more likely to conduct the tests (*p* < 0.001). The interpretation of these tests was mostly conducted by doctors in about two thirds of cases.

The majority of respondents believed that rapid tests were crucial for diagnosis and treatment, with most also agreeing that these tests could reduce unnecessary antibiotic use. However, there was a notable difference in perception between first- and second-level hospitals, where most agreed on their utility, and third-level hospitals, where a majority did not believe rapid tests would reduce antibiotic use (*p* < 0.05).

When comparing regional differences between the North and Middle Italy regions and the South and Isles regions, several distinctions emerged. A higher proportion of hospitals in the South and Isles had specific pediatric access (56.25% vs. 31.71% in the North and Middle regions, *p* < 0.05). The North and Middle regions were more likely to have disinfection protocols and frequent air changes, as well as brochures available in multiple languages (*p* < 0.01). In contrast, the South and Isles regions had more short-stay observation rooms with private bathrooms (*p* < 0.05).

## 4. Discussion

To our knowledge, this is one of the first national surveys that comprehensively investigates the measures adopted in pediatric EDs to prevent the spread of infectious diseases. The survey uniquely focuses on the tools available to inform both patients and healthcare workers about infections and the measures in place to prevent their transmission, including the availability of hand sanitizer, automatic doors, electronic sinks, PPE, and disinfection protocols. Moreover, this study offers a cross-sectional view of how different Italian hospitals operate regarding pediatric care, highlighting regional differences and variations across different hospital types.

Our findings are significant given the limited existing literature on the topic, particularly in pediatric settings. Previous reviews have largely focused on adult populations outside Italy and have not provided the detailed analysis seen in our study [15]. The existing literature primarily emphasizes general recommendations for infection control in EDs, such as the importance of hand hygiene, transmission-based precautions, environmental cleaning, and the proper reprocessing of reusable medical devices [16,17,18]. However, these studies often lack detailed data on the actual availability and implementation of these practices. Most of the evidence available is observational, with considerable variability in methods, outcomes, and reporting standards [19].

Our survey revealed that while nearly half of the respondents were from Northern Italy, and most were over 35 years old, a significant proportion worked in general hospitals or I level assistance hospitals. Only a small percentage represented III level hospitals. One of the key findings is that less than half of non-pediatric hospitals have separate access for children in their EDs, which raises concerns about the potential exposure of children to pathogens from adult patients during their stay in shared spaces.

The reported lack of planning, insufficient simulation training, inappropriate use of PPE, and inadequate isolation facilities prior to the pandemic underscore critical areas of vulnerability in pediatric EDs [20]. These deficiencies highlight the urgent need for structured preparedness programs, including regular training, resource allocation, and infrastructure improvements, to mitigate risks and enhance resilience against future infectious disease outbreaks.

In terms of infection control measures, masks and hand sanitizer were provided across all settings before patients entered visiting rooms. However, I level assistance hospitals were more likely to ensure social distancing and provide informational materials on infectious diseases. This could be due to the smaller number of pediatric accesses in these settings, allowing for more individualized care.

Interestingly, we found no significant differences among the three hospital levels regarding the presence of disinfection protocols and their application frequency. The absence of toys in waiting rooms, likely a post-COVID-19 precaution, was another common feature across most settings.

A notable difference emerged concerning the availability of automatic doors, with most III level assistance hospitals having them, while I level assistance hospitals mostly did not. This likely reflects the higher volume of general patient access in the latter.

Regarding rapid diagnostic tests, our study showed that while most settings had urine sticks, blood gas analyses, and SARS-CoV-2 antigen tests available, there was a notable focus on COVID-19 testing over other respiratory pathogens, despite the latter often having a more significant impact on pediatric populations. This emphasis on COVID-19 is consistent with global practices aimed at controlling the virus’s spread in healthcare settings [21]. However, studies have pointed out the importance of testing for a broader range of respiratory viruses in EDs to prevent nosocomial infections [22,23]. Most rapid assays, including those for non-COVID respiratory viruses, demonstrate adequate analytical performance, but more high-quality studies are needed to assess their impact fully. These assays must be appropriately integrated into ED workflows, considering local constraints and priorities [24].

Our analysis also highlighted regional disparities. In North–Middle Italy, more nurses responded to the survey, and there was a greater availability of brochures in multiple languages and better training for healthcare workers in PPE usage. In contrast, South–Isles regions were more likely to have separate pediatric access in EDs and a higher number of short-stay observation rooms with private bathrooms.

Overall, the differences between hospital levels and regional settings underline the need for tailored infection control measures that consider the specific challenges and resources of each setting. Implementing local checklists for infection control, as recommended by some professional societies, could be a practical next step to enhance infection prevention efforts in EDs across Italy [25,26,27,28]. A limitation of our study is that the data were self-reported by respondents, which may not allow the full capture of the actual practices and their effectiveness in reducing infection transmission. Another limitation of the study lies in the construction of the survey questions, which measured only a selected subset of parameters relevant to infection transmission in ED settings, potentially overlooking other critical factors that influence infection control practices.

## 5. Conclusions

Our findings indicate that while basic tools for managing infectious diseases (such as hand hygiene sanitizer, dedicated rooms for tests and visits, and disinfection protocols) are widely available, there are significant areas for improvement. These include ensuring social distancing in all settings and establishing dedicated pediatric pathways separate from adult patient flows. Future studies should evaluate key elements of hand hygiene infrastructure in pediatric EDs, including the number of handwash basins per facility, the availability of electronic sinks, the type of soap provided, available hand-drying methods, and whether the informational materials emphasize proper hand hygiene practices, as these factors are essential for improving infection prevention measures and compliance.

The survey reflects a generally good state of infection control in Italian EDs but highlights disparities that need addressing, particularly in preparation for future pandemics. Strengthening these areas will be crucial in enhancing the resilience of healthcare systems and ensuring the safety of both patients and healthcare workers in the face of emerging infectious threats.

## Figures and Tables

**Table 1 jcm-13-07762-t001:** General characteristics of respondents and their centers.

Characteristic	N (%)
**Age (Years)**	
25–35	9 (11.5)
35–45	32 (40.0)
>45	39 (48.7)
**Region of Italy**	
North	38 (47.5)
Middle	9 (11.3)
South	17 (21.5)
Isles	16 (20.0)
**Type of Hospital**	
Pediatric hospital	7 (8.8)
24 h pediatric assistance hospital	39 (48.8)
I level Emergency Department	9 (11.3)
II level Emergency Department	20 (25.0)
Scientific hospitalization and treatment institution	5 (6.3)
**Work Role**	
Other	1 (1.2)
Nurse	13 (16.3)
Resident	1 (1.2)
Medical doctor	65 (81.3)
**Annual Pediatric Emergency Department Visits**	
<10,000	28 (35)
10,000–20,000	19 (23.75)
20,000–30,000	15 (18.75)
>30,000	18 (22.5)
**Specific Access for Children (Non-Pediatric Hospitals)**	
Separate pediatric access	31/73 (42.5)
Pediatric fast track	44/73 (60.3)

**Table 2 jcm-13-07762-t002:** Characteristics of waiting and visiting rooms by hospital setting.

Characteristic	First Level	Second Level	Third Level	*p*-Value
	(N = 48)	(N = 20)	(N = 12)	
**Waiting Rooms**				
Masks available	28 (58.33)	14 (70.00)	8 (66.67)	0.721
Hand sanitizer available	43 (89.58)	17 (85.00)	11 (91.67)	0.879
Social distancing ensured	31 (64.58)	13 (65.00)	5 (41.67)	0.353
Informational material available	21 (43.75)	4 (20.00)	4 (33.33)	0.184
**Visit Rooms**				
Sink available	42 (87.50)	18 (90.00)	11 (91.67)	0.99
Workers wear protective gear	48 (100.00)	20 (100.00)	12 (100.00)	-
Workers trained in PPE use	45 (93.75)	18 (90.00)	11 (91.67)	0.846
Parents informed of infection risk	29 (60.42)	14 (70.00)	8 (66.67)	0.846
**Short-Stay Observation Rooms**				
Single rooms available	17 (35.42)	4 (20.00)	2 (16.67)	0.32
Private bathroom in rooms	40 (83.33)	15 (75.00)	4 (33.33)	<0.01
Infectious patients separated	32 (66.67)	9 (45.00)	9 (75.00)	0.17

**Table 3 jcm-13-07762-t003:** Disinfection protocols by hospital setting.

Characteristic	First Level	Second Level	Third Level	*p*-Value
	(N = 48)	(N = 20)	(N = 12)	
Disinfection protocols present	44 (91.67)	17 (85.00)	11 (91.67)	0.767
Disinfection frequency (3–5 times/day)	19 (39.58)	8 (40.00)	9 (75.00)	0.127
Toys available	22 (45.83)	7 (35.00)	9 (75.00)	0.782
Automatic door entrance	11 (22.92)	7 (35.00)	7 (58.33)	0.064

**Table 4 jcm-13-07762-t004:** Rapid test usage by hospital setting.

Characteristic	First Level	Second Level	Third Level	*p*-Value
	(N = 48)	(N = 20)	(N = 12)	
Urine stick available	45 (93.75)	20 (100.00)	12 (100.00)	0.73
Rapid blood count available	25 (52.08)	10 (50.00)	4 (33.33)	0.57
Rapid antigenic test for SARS-CoV-2 available	41 (85.42)	18 (90.00)	9 (75.00)	0.54
Rapid tests performed in visit room	23 (47.92)	5 (25.00)	8 (66.67)	<0.05
Rapid tests performed by doctors	4 (8.33)	1 (5.00)	4 (33.33)	<0.001
Rapid tests determinant for diagnosis	40 (83.33)	14 (70.00)	9 (75.00)	0.51
Rapid tests reduce antibiotic use	44 (91.67)	17 (85.00)	0 (0.00)	0.73

## Data Availability

All the data are included in the manuscript.

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
