# Peer review of "Infectious Risk in Pediatric Emergency Departments in Italy: A Survey by the Italian Society for Pediatric Emergency and Urgent Medicine (SIMEUP) on Available Preventive and Diagnostic Tools"

_jcm, 2024, doi:10.3390/jcm13247762_

Round 1

Reviewer 1 Report

Comments and Suggestions for Authors

Title: Infectious Risk in Pediatric Emergency Departments in Italy: A 2 Survey by the Italian Society for Pediatric Emergency and Urgent Medicine (SIMEUP) on Available Preventive and Diagnostic Tools.

This empirical study aimed to evaluate the current practices and available tools for infection control in Pediatric EDs across Italy, focusing on the differences between various hospital types and regional settings.

The presentation of article was generally good with some interesting points made in the discussion. However there are some general comments for improvement:

In the Introduction/Rationale section, the authors mention: “These included the lack of early availability of documented contingency plans, insufficient simulation training, inappropriate use of personal protective equipment (PPE), and inadequate isolation facilities” (Lines 67-69).

In the results or discussion section I didn’t see any mention of simulation or staff training so was this element not surveyed? And if not, why?

In the Discussion section, it mentions “hand hygiene facilities” (Line 186), but in the Results section only the availability of hand sanitiser is mentioned. There is a difference between hand hygiene facilities and amenities. From the paper, it is not clear what HH facilities there were (or how they were defined) – i.e. how many wash-hand basins per ED (facility element), what type of soap if any was used at the wash-hand basins, what drying methods were available (amenities), did any of the informational material available focus on HH etc.,

Furthermore, Line 139 mentions “hand sanitizer” but Line 205 mentions “hand gel” – are these the same thing? If so, the same terminology should be consistent throughout the article. Similarly, in the Discussion section, the authors mention “hand hygiene facilities” (Line 186) and then in the Conclusion it mentions “hand hygiene stations” (Line 242) – are these also the same thing?

Author Response

This empirical study aimed to evaluate the current practices and available tools for infection control in Pediatric EDs across Italy, focusing on the differences between various hospital types and regional settings.

The presentation of article was generally good with some interesting points made in the discussion. However there are some general comments for improvement:

Re: Thank you very much for your positive evaluation. We improved the text following your suggestions.

In the Introduction/Rationale section, the authors mention: “These included the lack of early availability of documented contingency plans, insufficient simulation training, inappropriate use of personal protective equipment (PPE), and inadequate isolation facilities” (Lines 67-69). In the results or discussion section I didn’t see any mention of simulation or staff training so was this element not surveyed? And if not, why?

Re: The text has been improved according to your comment (pp. 4 and 7).

In the Discussion section, it mentions “hand hygiene facilities” (Line 186), but in the Results section only the availability of hand sanitiser is mentioned. There is a difference between hand hygiene facilities and amenities. From the paper, it is not clear what HH facilities there were (or how they were defined) – i.e. how many wash-hand basins per ED (facility element), what type of soap if any was used at the wash-hand basins, what drying methods were available (amenities), did any of the informational material available focus on HH etc.,

Re: The text has been clarified according to your comments (pp. 7-8).

Furthermore, Line 139 mentions “hand sanitizer” but Line 205 mentions “hand gel” – are these the same thing? If so, the same terminology should be consistent throughout the article. Similarly, in the Discussion section, the authors mention “hand hygiene facilities” (Line 186) and then in the Conclusion it mentions “hand hygiene stations” (Line 242) – are these also the same thing?

Re: The text has been corrected according to the suggestions.

Reviewer 2 Report

Comments and Suggestions for Authors

It would be relevant to know the number of surveys sent and what percentage the 80 surveys received represent.

Is the total number of paediatric emergency centres in Italy 119?

What percentage of centres are represented by these 80 surveys?

Author Response

Re: Thank you very much for your comments. We revised the manuscript according to your suggestions.

It would be relevant to know the number of surveys sent and what percentage the 80 surveys received represent.

Re: Clarified (pp. 1 and 3).

Is the total number of paediatric emergency centres in Italy 119?

Re: Yes, it is.

What percentage of centres are represented by these 80 surveys?

Re: Clarified (pp. 1 and 3).

Reviewer 3 Report

Comments and Suggestions for Authors

The paper aims to evaluate practices and tools useful in the prevention and control of infection spread in Pediatric Emergency Departments in Italy depending on the region. The topic is up-to-date, and the paper contributes to better understanding of current practices and facilities in Italian health care system. However, there are some concerns about the study methodology, that should be addressed before the publication. Below are my comments and suggestions. 

Methods 

  • Survey instrument design - lack of information about survey reliability and validity 

  • Survey administration method – in terms of target population, representative character of the study group, coverage error, for example, there is no information about the response rate (how many e-mail addresses were on the mailing list, how many surveys were completed, how many were excluded from the final analysis) 

  • As there were collected data regarding region and town, it should be stated, if analyzed 80 responses were from different EDs (80 of 119), or were there multiple responses from the same center? 

  • Could the authors provide a response rate for each region? How many Pediatric EDs are in each region (North/ Middle/ South / Isles), if the SIMEUP members represent all regions similar percentage? - the information would be important in terms of data comparison and analysis of regional differences, because may affect the interpretation of the results  

Results 

Table 1 - Pediatric Access Numbers – in what timeframe, per year? - should be clarified 

Discussion 

186: hand hygiene facilities (...) electronic sinks – the survey included one question about presence of a sink in the visiting room and one question about hand disinfectant in waiting room, no questions about electronic sinks or hand disinfectants in visiting room 

237-238: in Limitations, I would suggest addressing not only self-report bias, but also other methodological problems and biases, as well as construct of the survey questions and measurement only selected parameters that might play a role in transmission of infection diseases in ED settings.

Author Response

The paper aims to evaluate practices and tools useful in the prevention and control of infection spread in Pediatric Emergency Departments in Italy depending on the region. The topic is up-to-date, and the paper contributes to better understanding of current practices and facilities in Italian health care system. However, there are some concerns about the study methodology, that should be addressed before the publication. Below are my comments and suggestions.

Re: Thank you for your comments. We revised the text according to your recommendations.

Methods

Survey instrument design - lack of information about survey reliability and validity

Re: Clarified (p. 3).

Survey administration method – in terms of target population, representative character of the study group, coverage error, for example, there is no information about the response rate (how many e-mail addresses were on the mailing list, how many surveys were completed, how many were excluded from the final analysis)

Re: Clarified (p. 3).

As there were collected data regarding region and town, it should be stated, if analyzed 80 responses were from different EDs (80 of 119), or were there multiple responses from the same center?

Re: Only one questionnaire per ED was evaluated (p. 3).

Could the authors provide a response rate for each region? How many Pediatric EDs are in each region (North/ Middle/ South / Isles), if the SIMEUP members represent all regions similar percentage? - the information would be important in terms of data comparison and analysis of regional differences, because may affect the interpretation of the results 

Re: Clarified (p. 3).

Results

Table 1 - Pediatric Access Numbers – in what timeframe, per year? - should be clarified

Re: Clarified (p. 4).

Discussion

186: hand hygiene facilities (...) electronic sinks – the survey included one question about presence of a sink in the visiting room and one question about hand disinfectant in waiting room, no questions about electronic sinks or hand disinfectants in visiting room

Re: A comment has been added as suggested (p. 9).  

237-238: in Limitations, I would suggest addressing not only self-report bias, but also other methodological problems and biases, as well as construct of the survey questions and measurement only selected parameters that might play a role in transmission of infection diseases in ED settings.

Re: Added (p. 8).